# Bacterial Pathogenesis in Various Fish Diseases: Recent Advances and Specific Challenges in Vaccine Development

**DOI:** 10.3390/vaccines11020470

**Published:** 2023-02-17

**Authors:** Aadil Ahmed Irshath, Anand Prem Rajan, Sugumar Vimal, Vasantha-Srinivasan Prabhakaran, Raja Ganesan

**Affiliations:** 1Department of Biomedical Sciences, School of Bio Sciences and Technology, Vellore Institute of Technology (VIT), Vellore 632 014, Tamil Nadu, India; 2Department of Biochemistry, Saveetha Medical College & Hospital, Saveetha Institute of Medical and Technical Sciences (SIMATS), Thandalam, Chennai 600 077, Tamilnadu, India; 3Department of Bioinformatics, Saveetha School of Engineering, Saveetha Institute of Medical and Technical Sciences (SIMATS), Chennai 600 077, Tamilnadu, India; 4Institute for Liver and Digestive Diseases, College of Medicine, Hallym University, Chuncheon 24253, Republic of Korea

**Keywords:** fish, aquaculture, bacterial pathogens, immunity, vaccines

## Abstract

Aquaculture is a fast-growing food sector but is plagued by a plethora of bacterial pathogens that infect fish. The rearing of fish at high population densities in aquaculture facilities makes them highly susceptible to disease outbreaks, which can cause significant economic loss. Thus, immunity development in fish through vaccination against various pathogens of economically important aquaculture species has been extensively studied and has been largely accepted as a reliable method for preventing infections. Vaccination studies in aquaculture systems are strategically associated with the economically and environmentally sustainable management of aquaculture production worldwide. Historically, most licensed fish vaccines have been developed as inactivated pathogens combined with adjuvants and provided via immersion or injection. In comparison, live vaccines can simulate a whole pathogenic illness and elicit a strong immune response, making them better suited for oral or immersion-based therapy methods to control diseases. Advanced approaches in vaccine development involve targeting specific pathogenic components, including the use of recombinant genes and proteins. Vaccines produced using these techniques, some of which are currently commercially available, appear to elicit and promote higher levels of immunity than conventional fish vaccines. These technological advancements are promising for developing sustainable production processes for commercially important aquatic species. In this review, we explore the multitude of studies on fish bacterial pathogens undertaken in the last decade as well as the recent advances in vaccine development for aquaculture.

## 1. Introduction

There have been deliberate discussions regarding sustainability in aquaculture worldwide over the last two decades [1]. Aquaculture is currently the world’s fastest-growing food sector [2], with a global production of 85.3 million tons in 2019. It contributes significantly to nutrition and food security, particularly in some of the most food-insecure regions, while supporting the livelihood of several million people worldwide. In 2018, aquaculture contributed to 46% of global fish production, with Asia dominating 80% of global aquaculture production by quantity and economic value. The United Nations Food and Agricultural Organization (FAO) estimated that a 70% increase in the world’s food and feed supply will be required to maintain the expanding human population in 2050. As a result of human population expansion and the increasing wealthy lifestyle of people in the Asia-Pacific region, the demand for aquaculture is expected to rise by 30% by 2030 [3].

Currently, aquaculture production is the fastest expanding animal food sector in the world [4]. In 2019, aquaculture production totaled 85.3 million tons, up 3.7% over 2018, with China (48.2 million tons), India (7.8 million tons), Indonesia (6.0 million tons), Vietnam (4.4 million tons), Bangladesh (2.5 million tons), Egypt (1.6 million tons), Norway (1.5 million tons), Chile (1.4 million tons), Myanmar (1.1 million tons), and Thailand (1 million tons) were the top ten aquaculture producers in 2019 (Figure 1) [5]. In the world of aquaculture, inland finfish culture was the most important sector. In 2019, 56.3 million tons of finfish (66.0%), 17.6 million tons of mollusks (20.6%), 10.5 million tons of crustaceans (12.3%), and 977 thousand tons of other aquatic animal species were produced in aquaculture around the world. Between 2011 and 2015, the global aquaculture production of aquatic animals grew at a pace of 5.0 percent each year on average. During the period 2016–2019, the yearly growth rate slowed to an average of 3.7 percent. Aquaculture has progressively increased its share of total aquatic animal output from capture and aquaculture combined from 39.9% in 2010 to 48.0% in 2019. Fish accounted for roughly 17.3% of the global population’s animal protein intake and 6.8% of all proteins taken in 2017, with global per capita consumption of fish estimated at 20.3 kg. Fish provides around 3.3 billion people, with nearly 20% of their average per capita diet on animal protein, and 5.6 billion people with 10% of such protein. By the year 2025, total global fish production is predicted to reach 196 million tons (Mt), with aquaculture expected to overtake total catch fisheries production [2]. In the last three decades, from 1990–2018, it showed 527% growth reached by producing 82 million tons for the estimated value of 250 billion USD of its first sale [4,5].

Applications of science and the implementation of advanced technologies in aquaculture development have accelerated aquaculture development during the past half-century [4]. Utilizing contemporary biotechnological methods to increase fish production has the potential to significantly increase fish quality and quantity in aquaculture systems while also meeting demand [2]. Aquaculture is more diverse than other agricultural industries in terms of species, food, culture processes, products, disease conditions, and ecosystems [3]. Majorly the sector focuses on ecosystem-based management and production system design and encourages sustainable production [4]. Scientific and technological advances have benefited almost every aspect of aquaculture. A lot of technologies have contributed significantly to the production of aquaculture. For example, improved reproductive technologies have enabled people to close the life cycles of aquaculture species, which provides for species diversification in aquaculture [1,5]. Selective breeding with the help of quantitative genetics has substantially improved traits of commercial importance in over 60 aquaculture species [6,7]. Selection based on genomic data (genomic selection) has the tremendous potential to alter genetic improvement programs and production systems within the aquaculture industries [8]. Improved feed formulations based on the nutritional requirements of each fish species have improved feed conversion rate (FCR) and reduced feed cost [9]. Technologies for disease management have reduced the occurrence of diseases in aquaculture.

## 2. Aquaculture Diseases

Infections in fish leading to disease outbreaks are a major concern for the aquaculture sector because they can result in significant economic damage owing to morbidity and death. The high fish-rearing densities currently used in aquaculture enable the transfer and spread of pathogenic microorganisms and are often a primary cause of such catastrophic outbreaks [4]. Intensive farming practices exert huge stresses on cultured aquatic species, compromising their innate immune defenses against various disease-causing bacterial and viral pathogens. Adequate husbandry and overall management, including biosecurity, nutrition genetics, system management, and water quality, are crucial for aquaculture production in all intensive culture farming practices, irrespective of whether individual or several species of fish are produced in dense populations [8]. In China, India, and Vietnam, fish diseases are estimated to contribute to more than 30% of the overall production loss [9]. Several bacterial and viral pathogens and parasites are opportunistic and occur in the environment or as asymptomatic carriers on some fish, which renders aquaculture facilities highly susceptible to disease outbreaks and hinders the development of an efficient, cost-effective, and stable aquaculture process [10]. The appearance and progression of fish disease are determined by the relationship between the pathogen, host, and environment. Stressful conditions, including high population density, change in temperature, and hypoxia, can hasten the spread of pathogenic bacteria and result in major disease outbreaks [11]. Thus, multidisciplinary studies on the characteristics of potential fish pathogens, the biology of the fish hosts, and an adequate understanding of the global environmental factors affecting are important to investigate appropriate measures for the prevention and control of the major diseases limiting fish production in aquaculture.

## 3. Bacterial Pathogens of Fish

Several bacterial infections in fish species, including Aeromonas septicemia [12], Edwardsiellosis [13], Columnaris [14], Streptococcosis [15], and vibriosis [16] have been reported in the aquaculture sector [17]. Nevertheless, a few of these pathogens are found to be highly responsible for the majority of global economic losses in aquaculture production [18]. Bacterial species responsible for disease outbreaks in different fish species are mentioned in Table 1. *Aeromonas* spp. are among the most common types of bacterial pathogens in numerous fish species that occur in freshwater and tropical environments and cause bacterial hemorrhage in cultured fishes [19]. *Aeromonas salmonicida* is one of the oldest known fish pathogens that occurs worldwide in both fresh and marine waters aquaculture regions and is associated with skin ulceration and hemorrhages found as recurrent clinical symptoms of infection [20,21].

*Aeromonas hydrophila* predominantly causes disease outbreaks in cultured freshwater fish, contributing to global food insecurity and economic losses in aquaculture production [79]. *A. hydrophila* infections are associated with various symptoms such as hemorrhagic septicemia, edema, epizootic ulcerative syndrome (EUS), hemorrhagic enteritis, and red body disease and affect different cultivable finfish species, including common carps, goldfish, eel, catfish and tilapia fishes [12].

*Edwardsiella tarda* is another globally occurring fish pathogen isolated from both fresh and seawater and also from the intestines of normal aquatic animals. It is an intracellular pathogen that affects a wide variety of hosts, producing illnesses not only in fishes but also in amphibians, reptiles, birds, and mammals worldwide [30].

*Yersinia ruckeri* is found across aquaculture facilities in North and South America, Europe, and South Africa. It causes significant economic losses in salmon aquaculture production in countries such as Norway, Chile, Australia, and Scotland, where it shows the ability to survive in nutrient-limiting environments, inside or outside the host, and facilitates the transmission of infections [34].

*Piscirickettsia salmonis*, a non-motile obligate intracellular gram-negative bacterium, causes Salmon Rickettsia Syndrome (SRS), also known as piscirickettsiosis. SRS corresponds to an aggressive infectious disease affecting the economy of global salmon production. *P. salmonis* has been unequivocally declared as the agent responsible for dramatic economic losses suffered by the Chilean salmon industry in the last decade, and it has also impacted aquaculture production in western Canada, Norway, and Ireland regions. [38,80].

In *Flavobacterium*, three species have been found to infect freshwater and wild fish populations globally: *F. columnare*, causing columnaris disease, *F. branchiophilum* causing bacterial gill disease, and *F. psychrophilum,* causing bacterial cold-water disease. These species are associated with one of the widest host and geographic ranges among deadly fish pathogens [41,42].

*Pseudomonas anguilliseptica* is an opportunistic pathogen affecting a variety of fish species in marine and brackish water aquaculture production around the world. Originally described as the causative agent of red spot disease in Japanese eel culture, it has been since isolated in different countries from a variety of cultured and wild fish species such as European eel, black sea bream, ayu, Atlantic salmon, sea trout, rainbow trout, whitefish, Baltic herring, striped jack, and orange-spotted grouper [46,47].

Vibriosis is another major hindrance to fishery production that affects a wide range of aquaculture animals globally. The main widespread causative agents of vibriosis include *Vibrio harveyi* and *V. anguillarum*, which are halophilic bacteria existing in aquatic and marine environments and infect a large number of economically important fishes. *V. anguillarum* causes highly fatal hemorrhagic septicemia in many kinds of fish species, including high value Atlantic salmon (*Salmo salar*), Rainbow trout (*Oncorhynchus mykiss*), and Japanese seaperch (*Lateolabrax japonicus*). It often leads to the large-scale death of fish and consequent substantial economic losses in aquaculture production [49,81]. Vibriosis also affects groupers, a popular carnivorous fish species found in the Atlantic Ocean and Mediterranean Sea’s tropical and subtropical seas, with a strong market demand in many nation’s consumers. *V. carchariae*, *V. alginolyticus*, *V. harveyi*, and *V. parahaemolyticus* are additional examples of Vibrio fish pathogens [55,56].

*Moritella viscosa* is the prime causative agent of winter-ulcer disease, affecting fish reared in marine waters at temperatures below 8 °C. The disease outbreaks caused by *M. viscosa* are primarily experienced in salmonid farming, where infected fish develop extensive ulcer lesions in external tissues and internal pathological changes [57,59].

Tenacibaculosis is a crucial bacterial disease that affects a major number of marine fish species, causing heavy losses for the aquaculture industry worldwide. The disease is caused by *Tenacibaculum maritimum* with characteristic symptoms of gross lesions on the body surface of fish such as ulcers, necrosis, eroded mouth, frayed fins, tail rots, and occasionally necrosis on the gills and eyes of the infected regions [61,62].

*Lactococcus garvieae* are gram-positive, hemolytic, chain-forming cocci that have been linked to fatal hemorrhagic septicemia and also cause meningoencephalitis in fish species and other animals. This bacterium is an emerging fish pathogen that affects a wide range of fish in freshwater and marine habitats and causes significant economic losses in aquaculture in the Mediterranean region, Japan, Europe, Southeast Asian countries, and North America. *L. garvieae* was first isolated from clinical samples of bovine mastitis in the UK region and then simultaneously from yellowtail (*Seriola quinqueradiata*) in Japan. Warm water lactococcosis, caused by *L. garvieae* during the summer months when water temperatures cross above 21 °C, has developed as a major deadly illness of farmed production of rainbow trout over the last few decades [82,83].

Streptococcal infections in fish, which were first reported in rainbow trout in Japan in 1958, have caused significant mortality in both wild and farmed fish, resulting in considerable economic losses to the aquaculture industry. The microbial species acting as etiological agents of streptococcosis in fish include *Lactococcus garvieae*, *L. piscium*, *Streptococcus agalactiae*, *S. iniae*, *S. parauberis*, and *Vagococcus salmoninarum*. *Streptococcus parauberis* was first identified as a fish pathogen after an outbreak in cultured turbot (*Scophthalmus maximus*) in the regions of Spain. It is also responsible for streptococcosis in olive flounder fish. *Streptococcus phocae* is a beta-hemolytic species belonging to Lancefield groups C, F, or G isolates. The species was first isolated and described in Norway from lung specimens from harbor seals suffering from pneumonia disease. This opportunistic pathogenic bacterium has been identified among several species of pinnipeds like the Cape fur seal, the Caspian seal, the spotted seal, the harbor seal, and sea lion species of different regions [73,74,84].

Mycobacteriosis is a chronic and frequently fatal disease that affects a variety of cultured and wild fish species around the world. Many *Mycobacterium* spp. have been recovered from diseased fish, with *M. marinum* being the most important due to its broad host range, economic effect on aquaculture globally, and zoonotic potential. Mycobacteriosis has caused severe damage to intensive farming and the ornamental trade, and there is currently no viable therapy other than depopulation and facility of the disinfection process. Fish mycobacteriosis is a chronic progressive disease caused by ubiquitous acid-fast bacilli, identified as nontuberculous mycobacteria (NTM). NTM are classified into slow (including *M. marinum*) and rapidly growing mycobacteria. *M. marinum*, *M. fortuitum*, and *M. chelonae* are among the prominently identified NTM species associated with fish mycobacteriosis disease. Piscine mycobacteriosis is a deadly disease commonly affecting marine, brackish, and freshwater fish and infecting approximately 200 species of marine and freshwater fish in a wide region extending from the subarctic zone to the tropical one. This disease also infects tropical aquarium fish and is considered to cause mortality and morbidity in free-living fishes [76,78,85].

## 4. Fish Vaccines—An Introduction

Fish infections continue to be a serious economic issue in commercial aquaculture around the world, despite many initiatives to develop new therapies [86]. Although antibiotics or chemotherapeutics may be used to treat fish disease, these are associated with obvious disadvantages such as drug resistance and safety concerns of consumers and the environment [87]. Vaccination is an effective technique to prevent a large variety of bacterial and viral infections and contributes to the environmental, social, and economic sustainability of aquaculture production globally [88]. Since the initial reports in the 1940s, several vaccines have been developed that have greatly reduced the impact of loss caused by bacterial and viral infections in fish [89,90]. Millions of fish are currently vaccinated each year, and there has been a shift away from using various antibiotics and toward immunization in different parts of the world [91].

A component either contained in or produced from the fish pathogen is used as an antigen to develop the vaccine [88,92]. This component will be involved in the activation of the innate or adaptive immune responses of the fish in response to a specific microbial infection. Over 100,000 research reports on fish vaccine development have been published in the last two decades, as well as several reviews on the history, developments, types, and routes of administration, and the opportunities and challenges of producing fish vaccines have been studied elaborately [93]. Many studies have summarized the importance of using adjuvants and immunostimulants in boosting the immune response of fish vaccinations, as well as delivery strategies [94,95]. Alternative vaccine administration techniques (other than injection) and the protective efficacies of old and promising new-generation adjuvants are being explored and evaluated.

## 5. Commercial Fish Vaccines

Several inactivated, live-attenuated, and DNA vaccines have been developed and are currently applied in large-scale fish farming operations. The first successful available commercial bacterial vaccine was developed against enteric redmouth disease and vibriosis and was introduced in the United States in the late 1970s. It was developed based on whole-cell inactivation and administrated through immersion methods [96,97]. Since 1990 the global development of fish vaccines has followed a path similar to that of human and veterinary vaccines, with extensive interactions between research and development, pharmaceutical industries, and regulatory bodies of concerned geographical regions. The major fish vaccine producers include Novartis Animal Health (Switzerland), Intervet International (The Netherlands), Pharmaq (Norway), Bayer Animal Health (Bayotek)/Microtek, Inc. (Germany/Canada), and Schering-Plough Animal Health (USA). The global commercial market for these companies is dominated by salmon and trout aquaculture productions [96].

There is a need for a comprehensive assessment of the current state of the fish vaccine sector due to the emergence of new vaccination technology developments. Over 26 licensed fish vaccines are available for use in a different range of fish species worldwide (Table 2). Most of the developed vaccines have been licensed for use in a number of aquaculture species by the United States Department of Agriculture (USDA) and are mainly prepared using traditional production methods that involve the cultivation of specific targeted pathogens [98,99]. According to the USDA, vaccines are currently provided to 77 types of fish against more than 22 types of different bacterial and six viral pathogenic specie [100]. Various countries, including Japan and Korea, have licensed and commercialized their fish vaccines [101,102]. In Japan, nine pharmaceutical industries produce fish vaccines for the Japanese market, with 29 vaccine formulations approved since 2018. Vaccines against eight bacterial species and two viral species have been approved and are in use for more than 13 types of fish species [101]. In Korea, 29 vaccines for ten types of fish pathogens are approved and commercially available [102].

## 6. Recent Studies on Fish Vaccines Development

Although many bacterial vaccines are available for commercial use in aquaculture productions, effective vaccines for many bacterial diseases have yet to be developed and produced [103]. Advances in molecular biology, biotechnology, and reverse vaccinology have permitted the production of several forms of vaccinations, including subunit vaccines, plasmid DNA vaccines, recombinant live vector vaccines, and recombinant protein vaccines, which have been experimentally tested in fish and have been approved for commercialization [79].

Most early in tradition, the vaccine trials were focused on killed vaccines. The first fish vaccine reported was a killed *Aeromonas salmonicida* oral vaccine of cutthroat trout *Oncorhynchus clarkii* [20,104]. The first available licensed commercial vaccine for fish was a killed vaccine of *Yersinia ruckeri* administrated by immersion methods against enteric redmouth disease [105]. With the success of this vaccine, formalin-killed immersion vaccines were developed for vibriosis of trout and salmon. Earlier salmonid vaccines were delivered by immersion and developed using the same technique for preventing bacterial infections in Atlantic salmon (*Salmo salar*) [98]. Biofilm vaccination is a highly effective strategy for reducing *A. hydrophila* infection; this contains both protective and non-protective proteins, which may result in a heterologous adaptive immune response in vaccinated fish [106].

Reverse vaccinology survey of potent antigenic target contents of specific pathogens were used to develop subunit vaccines against fish nocardiosis in the largemouth bass (*Micropterus salmoides,*), which demonstrated that the vaccines were highly promising for nocardial prophylaxis despite showing differential effects [107]. The efficacy of a vaccine against *Streptococcus agalactiae* in Nile tilapia was studied in the presence of salinity stress using serum antibody levels as a surrogate marker, as they may reliably correspond with the protective immunity elicited by fish vaccines. Because salinity stress can cause a variety of alterations, researchers gathered information on cell counts, cortisol levels, electrolytes, serum bactericidal activity, and fish survival after being exposed to *S. agalactiae* [108].

A multicomponent vaccine was demonstrated to protect trout against three relevant bacterial diseases (yersiniosis, furunculosis, and vibriosis) under various experimental conditions, indicating that the vaccine induces specific antibody responses to different bacterial antigens and regulates effective expressions of various genes involved in the immune response [109]. Similarly, a SagH gene-based DNA vaccine conferred an immunoprotective effect against *Streptococcus iniae* with a high relative percent survival (RPS) of 92.62% and 90.58% against *S. iniae* serotype I after 1 and 2 months, respectively. In addition, the vaccine conferred strong cross-protection against *S. iniae* serotype II and resulted in an RPS of 83.01% and 80.65% after 1 and 2 months, respectively [110]. An inactivated vaccine made of formalin-killed cells of *V. harveyi* with commercial adjuvant Montanide™ ISA 763 A VG conferred 75% RPS at four weeks post-vaccination [111].

There have been several reports of combination vaccinations containing multiple inactivated pathogens. After being challenged with *V. alginolyticus, V. parahaemolyticus*, and *Photobacterium damselae* subsp., a combination of these three inactivated bacterins demonstrated an RPS > 80% in cobia fish [112]. The immunization of Nile tilapia with formalin-killed cells of *S. agalactiae* or *S. iniae* provided protection against infection, with effective RPS values of 92.3% and 91.7%, respectively [113]. Immunization with an intracoelomic injection protected mice from a virulent wild-type strain of *S. iniae*, with RPS reaching 95.05% efficacy in Nile tilapia [114]. An inactivated recombinant vaccine encoding the cell wall surface anchored family protein of *S. agalactiae* was used to immunize the red hybrid tilapia (*Oreochromis* sp.). In serum, mucus, and gut lavage fluid samples, orally inoculated fish registered a strong and considerably high IgM antibody immune response with an efficacy of 70% RPS [115].

## 7. Conclusions

Large-scale reductions in the usage of antibiotics were brought on by effective fish vaccinations. But combining all factors that interfere with development to a ministration method remains the real issue in the fish vaccine. Despite several positive results in research and experimental trials with a moderate to high market potential for fish vaccines, there are only a few approved vaccines available on the market to protect against diseases in economically important fish. However, with recent advancements, multiple next-generation vaccine developments can be achieved against various infectious pathogens, especially bacteria, with more clearly defined adjuvants, microcarriers, and nanocarrier-based precisely targeted vaccines to produce higher protective immunity in cultured fish species, which may be available soon for the aquaculture sector. Research on vaccine formulations comprising the most suitable antigenic components, as well as field trial studies that corroborate laboratory findings, will aid in the development of a fish vaccine that is effective against the majority of bacterial infections. This will contribute to the sustainable growth of the economy and control the impact of environmental pollution caused by conventional antibiotics and chemical-based treatments.

## Figures and Tables

**Figure 1 vaccines-11-00470-f001:**
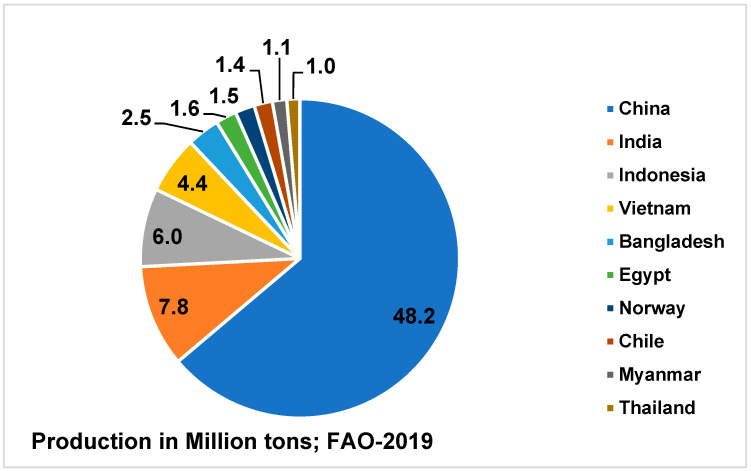
Global Aquaculture Production.

**Table 1 vaccines-11-00470-t001:** Bacterial Pathogens of Fishes.

Agents	Disease	Host Fish Targets	References
*Aeromonas salmonicida*	Furunculosis	trout, salmon, goldfish, koi, and a wide range of fish species	[20,22,23,24]
*Aeromonas hydrophila*	Motile Aeromonas septicemia (MAS), hemorrhagic septicemia, red-sore disease, ulcer disease, epizootic ulcerative syndrome (EUS)	tilapia, catfish, striped salmonid, non-salmonid fish, sturgeon, bass, and eel	[20,23,24,25]
*Edwardsiella ictaluri*	Enteric septicemia	Catfish and tilapia	[26,27,28,29]
*Edwardsiella tarda*	Edwardsiellosis	Salmon, carps, tilapia, catfish, striped bass, flounder, and yellowtail	[30,31,32]
*Yersinia ruckeri*	Enteric redmouth	Salmonids, eel, minnows, sturgeon, and crustaceans	[33,34,35,36]
*Piscirickettsia salmonis*	Piscirickettsiosis	Salmonids	[36,37,38,39]
*Flavobacterium psychrophilum*	Coldwater disease	Salmonids, carp, eel, tench, perch, ayu	[40,41,42]
*Flavobacterium columnare*	Columnaris disease	cyprinids, salmonids, silurids, eel, and sturgeon	[43,44,45]
*Pseudomonas anguilliseptica*	Pseudomonadiasis, winter disease	Sea bream, eel, turbot, and ayu	[46,47,48]
*Vibrio anguillarum*	Vibriosis	Salmonids, turbot, sea bass, striped bass, eel, ayu, cod, and red sea bream	[16,49,50]
*Vibrio salmonicida*	Vibriosis	Atlantic salmon, cod	[51,52,53]
*Vibrio carchariae*	Vibriosis, infectious gastroenteritis	Shark, abalone, red drum, sea bream, sea bass, cobia, and flounder	[54,55,56]
*Moritella viscosa*	Winter ulcer	Atlantic salmon	[57,58,59]
*Tenacibaculum maritimum*	Flexibacteriosis	Turbot, salmonids, sole, sea bass, gilthead sea bream, red sea bream, and flounder	[60,61,62]
*Lactococcus garvieae*	Streptococcosis or lactococcosis	Yellowtail, rainbow trout, and eel	[63,64,65,66]
*Streptococcus iniae*	Streptococcosis	Adriatic sturgeon, rainbow trout	[67,68,69]
*Streptococcus parauberis*	Streptococcosis	Turbot	[70,71,72]
*Streptococcus phocae*	Streptococcosis	Atlantic salmon	[73,74,75]
*Mycobacterium marinum*	Mycobacteriosis	Sea bass, turbot, and Atlantic salmon	[76,77,78]

**Table 2 vaccines-11-00470-t002:** USDA Approved Bacterial Fish Vaccines.

Disease	Pathogen	Vaccine Type	Delivery Methods	Country/Region	Make
Vibriosis	*Vibrio anguillarum; Vibrio ordalii; Vibrio salmonicida*	Inactivated	IP or IMM	USA, Canada, Japan, Europe, Australia	Merck Animal Health
Furunculosis	*Aeromonas salmonicida,* subsp. *Salmonicida*	Inactivated	IP or IMM	USA, Canada, Chile, Europe, Australia	MSD Animal Health
Bacterial kidney disease (BKD)	*Renibacterium, salmoninarum*	Avirulent live culture	IP	Canada, Chile, USA	Renogen
Enteric septicemia of catfish (ESC)	*Edwarsiella ictaluri*	Inactivated	IP	Vietnam	Pharmaq
Columnaris disease	*Flavobacterium columnaris*	Attenuated	IMM	USA	Merck Animal Health
Pasteurellosis	*Pasteurella piscicida*	Inactivated	IMM	USA, Europe, Taiwan, Japan	Pharmaq AS
Lactococcosis	*Lactococcus garvieae*	Attenuated	IP	Spain	hipara
Streptococcus infections	*Streptococcus* spp.	Inactivated	IP	Taiwan Province of China, Japan, Brazil, Indonesia	Aquavac-vaccines
Salmonid rickettsial septicemia	*Piscirickettsia salmonis*	Inactivated	IP	Chile	Pharmaq
Motile Aeromonas septicemia (MAS)	*Aeromonas hydrophila*, *A. caviae*, *A. sobria*	Inactivated	IP	Asia, Europe, United States	Pharmaq
Wound Disease	*Moritella viscosa*	Inactivated	IP	Norway, UK, Ireland, Iceland	Pharmaq
Tenacibaculosis	*Tenacibaculum maritimum*	Inactivated	IP	Spain	hipara
Channel Catfish Septicemia	*Edwardsiella ictaluri*	Avirulent live culture	IMM	United States	AquaVac
Enteric Redmouth Disease	*Yersinia ruckeri*	Attenuated	IMM	United States	Elanco (Aqua Health)

## Data Availability

Not applicable.

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
