# Peer review of "Bacterial Pathogenesis in Various Fish Diseases: Recent Advances and Specific Challenges in Vaccine Development"

_vaccines, 2023, doi:10.3390/vaccines11020470_

Round 1

Reviewer 1 Report (New Reviewer)

La revisión de este tema podría ser de gran interés. Sin embargo, hay varias cuestiones que deben tenerse en cuenta.

The introduction is focused on reviewing the current situation of aquaculture worldwide when it should address the subject of the review and justify the reason for the review.

Must improve fluency in the English language. Sometimes the wording is confusing and occasionally there are sentences that seem incoherent

It would be convenient to include a section on the immune system of fish and how vaccines work.

Please find the detail of several observations below:

Lines

 Introduction

The introduction presents only aquaculture production data. It should be summarized considerably and present there the objective of the review with general ideas on the topic of review.

141 to 144

The diseases and mortality rate caused by these pathogens together constitute one of the widest host and geographic ranges of any pathogenic bacteria of fishes

requires improving the wording to make the sentence easier to understand

186 to 187

Streptococcus phocae

Streptococcus phocae

204

moralityand morbidity

mortality and morbidity

216 to 217

A content that serves as an antigen is either contained in or produced from fish pathogen to develop the vaccine

I do not understand what the authors mean but perhaps the intention is to indicate that: to create the vaccine, a substance that serves as an antigen is either synthesized from or contained in a fish pathogen.

219

lakh

It is an Indian English term.

The journal allows to use it?

312

witha

with a

102 to 109

Aeromonas spp. are found to be one of the most common and predominantly …….

Separate from the previous sentence with a period

Author Response

La revisión de este tema podría ser de gran interés. Sin embargo, hay varias cuestiones que deben tenerse en cuenta.

The introduction is focused on reviewing the current situation of aquaculture worldwide when it should address the subject of the review and justify the reason for the review.

Must improve fluency in the English language. Sometimes the wording is confusing and occasionally there are sentences that seem incoherent

It would be convenient to include a section on the immune system of fish and how vaccines work.

Please find the detail of several observations below:

Lines

 Part

Suggesstion

Changes done

 Introduction

The introduction presents only aquaculture production data. It should be summarized considerably and present there the objective of the review with general ideas on the topic of review.

Irrelevant data has been removed and shorted

141 to 144

The diseases and mortality rate caused by these pathogens together constitute one of the widest host and geographic ranges of any pathogenic bacteria of fishes

requires improving the wording to make the sentence easier to understand

Rectified

186 to 187

Streptococcus phocae

Streptococcus phocae

Rectified

204

moralityand morbidity

mortality and morbidity

Done

216 to 217

A content that serves as an antigen is either contained in or produced from fish pathogen to develop the vaccine

I do not understand what the authors mean but perhaps the intention is to indicate that: to create the vaccine, a substance that serves as an antigen is either synthesized from or contained in a fish pathogen.

Changed

219

lakh

It is an Indian English term.

The journal allows to use it?

Rectified

312

witha

with a

Done

102 to 109

Aeromonas spp. are found to be one of the most common and predominantly …….

Separate from the previous sentence with a period

Done

Reviewer 2 Report (Previous Reviewer 2)

The manuscript vaccines-2159868 describes in general the recent advances on some bacterial diseases and on vaccines on the global market. The proposed review does not bring great news on the present scientific panorama, even if it collects some data in general and briefly describes the main pathologies. In itself the paper is pleasantly usable even if it does not describe anything new.

Going to evaluate chapter by chapter, some inaccuracies are noticed:

line 107 – spp. to be written in normal font;

line 123 – when talking about Edwardsiella tarda start the paragraph at the beginning;

line 132 – same thing for Piscirickettsia salmonis;

line 138 - spp. to be written in normal font;

line 143 – the paragraph relating to Pseudomonas anguilliseptica starts again;

line 152 - spp. to be written in normal font;

line 166 – the paragraph relating to tenacibaculosis starts again;

line 186-187 – Streptococcus phocae should be in italic font;

line 193 - spp. to be writted in normal font;

line 199 – remove the space between prominently and identified;

table 2 – in the Forunculosis row, the pathogen must be correctly written with subsp. in normal font; same thing for Edwardsiella ictaluri; in the Pasteurellosis row the pathogen is misclassified; in the last line correct Yersinia;

line 651-652 – the reference needs to be formatted .

Taken as a whole, the manuscript must be corrected for any inaccuracies identified; however, the publication process can continue after minor revision.

Author Response

Comments and Suggestions for Authors

The manuscript vaccines-2159868 describes in general the recent advances on some bacterial diseases and on vaccines on the global market. The proposed review does not bring great news on the present scientific panorama, even if it collects some data in general and briefly describes the main pathologies. In itself the paper is pleasantly usable even if it does not describe anything new.

Going to evaluate chapter by chapter, some inaccuracies are noticed:

line 107 – spp. to be written in normal font;

line 123 – when talking about Edwardsiella tarda start the paragraph at the beginning;

line 132 – same thing for Piscirickettsia salmonis;

line 138 - spp. to be written in normal font;

line 143 – the paragraph relating to Pseudomonas anguilliseptica starts again;

line 152 - spp. to be written in normal font;

line 166 – the paragraph relating to tenacibaculosis starts again;

line 186-187 – Streptococcus phocae should be in italic font;

line 193 - spp. to be writted in normal font;

line 199 – remove the space between prominently and identified;

table 2 – in the Forunculosis row, the pathogen must be correctly written with subsp. in normal font; same thing for Edwardsiella ictaluri; in the Pasteurellosis row the pathogen is misclassified; in the last line correct Yersinia;

line 651-652 – the reference needs to be formatted .

Taken as a whole, the manuscript must be corrected for any inaccuracies identified; however, the publication process can continue after minor revision.

Authors Response

            This review article is drafted precisely based on the recent improvements in vaccine technology in use against bacterial diseases in fish. The given reviewer comments for changes have been considered and executed accordingly. Thank you for your constructive suggestions.

Round 2

Reviewer 1 Report (New Reviewer)

Although the manuscript has been improved, there are still some typing errors that must be addressed and some clarifications to be made, such as the use of the scientific name V. anguillarum when the current name has been changed to Listonella anguilarum ; the use of words like dropsy generally known today as edema.

Lines

 Part

Suggesstion

69

[4] [5].

[4,5]

83

[1], [5].

[1,5]

95

microorganismand

microorganism and

117

Nevertheless, a few These pathogens are found to be highly responsible for the majority of global economic losses in aquacultureproduce

of these

aquaculture production?

120

Aeromonas spp. are among the most commontypes of bacterial pathogens

common types

125

[20. [21].

[20,21].

128

causesthedisease

causes the disease

131

dropsy

Edema

161-172

Please note that the name Vibrio anguillarum was changed to Listonella (syn. Vibrio) anguillarum. Why the authors prefer to use the name V. anguillarum?

174

The Disease

The disease

175

M.viscosa

M. viscosa

183

infishe

in fish

185

andcausing

and causing

187

L. garvieae was first isolated from clinical samples of mastitis in the UK

mastitis in fish? Please explain

Author Response

Reviewer 1

Comments and Suggestions for Authors

Although the manuscript has been improved, there are still some typing errors that must be addressed and some clarifications to be made, such as the use of the scientific name V. anguillarum when the current name has been changed to Listonella anguilarum ; the use of words like dropsy generally known today as edema.

Answer:

            Thank you for your constructive comments. All the suggestions and the changes mentioned by the reviewer is clarified and updated in the manuscript and highlighted here.

Lines

 Part

Suggesstion

Changes

69

[4] [5].

[4,5]

Done

83

[1], [5].

[1,5]

Done

95

microorganismand

microorganism and

Done

117

Nevertheless, a few These pathogens are found to be highly responsible for the majority of global economic losses in aquacultureproduce

of these

aquaculture production?

Done

120

Aeromonas spp. are among the most commontypes of bacterial pathogens

common types

Done

125

[20. [21].

[20,21].

Done

128

causesthedisease

causes the disease

Done

131

dropsy

Edema

Done

161-172

Please note that the name Vibrio anguillarum was changed to Listonella (syn. Vibrio) anguillarum. Why the authors prefer to use the name V. anguillarum?

Listonella is an older name, compare to Vibrio Ref: https://www.ncbi.nlm.nih.gov/pmc/articles/PMC6124819/#R4

174

The Disease

The disease

Done

175

M.viscosa

M. viscosa

Done

183

infishe

in fish

Done

185

andcausing

and causing

Done

187

L. garvieae was first isolated from clinical samples of mastitis in the UK

mastitis in fish? Please explain

Sentence mistake, initially it was isolated from bovine mastitis in UK and simultaneously from yellowtail fish from Japan – rectified.

This manuscript is a resubmission of an earlier submission. The following is a list of the peer review reports and author responses from that submission.

Round 1

Reviewer 1 Report

This review on fish vaccines is limited to vaccines agains bacterial fish pathogens. This may well be a reasonable limit, as the number of studies on vaccines against viral diseases of fish would greatly increase the number of references needed. Furthermore viral and bacterial vaccines are different in terms of modulation of the fish´immune defence.

The authors are right in their conclusion that despite the relatively big market, the number of approved vaccines is low. Here is a potential for improvements in fish health, economy, as well as environmental footprint of aquaculture.

Chapter 3 is useful and gives a good overview of bacterial diseases of fish. It might increase the value of the study if recent textbooks and reviews were added.

Chapter 4 would benefit from a few references to the historical information given on pioneering research in the 1940ies.

Chapter 5 "Commercial fish vaccines" should be somewhat extended. Fish vaccines are licensed in a range of countries, whereas for some reason only the USA is mentioned in the text. The US has good regulatory authorities, but is hardly any leader in aquaculture. Table 2 is useful, and could be extended and be more useful if more information is added on the limited number of vaccines (26) listed.  Details on which vaccine type each of them are, should be added.

Author Response

Reviewer 1:

Suggestion: This review on fish vaccines is limited to vaccines agains bacterial fish pathogens. This may well be a reasonable limit, as the number of studies on vaccines against viral diseases of fish would greatly increase the number of references needed. Furthermore, viral and bacterial vaccines are different in terms of modulation of the fish´immune defence.

Response: As mentioned there are many articles that are commonly published irrespective of pathogen types, but very less reviews are found specific for bacterial concern. This review might help fellow researchers, academicians and industrial specialists to learn precisely in short about bacterial fish diseases and its vaccines. 

Suggestion: Chapter 3 is useful and gives a good overview of bacterial diseases of fish. It might increase the value of the study if recent textbooks and reviews were added.

Respose: according to the given suggestion we have added some inputs from recently published book chapters and cited as well.

Suggestion: Chapter 4 would benefit from a few references to the historical information given on pioneering research in the 1940ies.

Response:  as the contents are precise to the recent developments historical inputs have also been mentioned and cited accordingly.

 Suggestion: Chapter 5 "Commercial fish vaccines" should be somewhat extended. Fish vaccines are licensed in a range of countries, whereas for some reason only the USA is mentioned in the text. The US has good regulatory authorities, but is hardly any leader in aquaculture. Table 2 is useful, and could be extended and be more useful if more information is added on the limited number of vaccines (26) listed.  Details on which vaccine type each of them is, should be added.

Response: As suggested the inputs have been given about the commercial vaccines of the countries like Japan and Korea and cited as well. The table have been restricted to US based vaccines as we have not found the data of the producing industries of the same.

Reviewer 2 Report

The manuscript vaccines-2013420 aims to track recent developments in vaccines against various bacterial pathogens affecting fish. Although written in a clear way, I do not find a linearity and a connection regarding the recent developments that have been conducted in recent years on this topic. This review lacks many ideas, especially original ones, to detach itself from the reviews already published in recent years in many scientific journals;  I think it is not a suitable manuscript to be published, because it lacks novelty, ideas to become interesting.

There are numerous inaccuracies in the text, such as line 105 where the species must be corrected.

In table 1 in the blot of Lactococcus garvieae, the rainbow trout is missing among the target species with the relative citations, as this gram-positive bacterium represents the main cause of death in this salmonid in farmed conditions. Streptococcus iniae, decidedly more important agent than S. parauberis, is also missing. It should be remembered that in line 133, the term spp is inserted in normal character. The pathology should be better highlighted on Moritella viscosa as it is among those most recently identified as a cause of pathology in fish. On line 183, italicize the scientific name and adjust the word Lancefield. On line 189 Mycobacterium goes in italics.

In Table 2 correct the name of disease Lactococcosis, correct the scientific name of the agent (Lactococcus garvieae), put spp. in normal font; insert ictaluri and ruckeri in lower case (last 2 lines) and correct the name of the etiological agent of Pasteurellosis.

 In line 247 correct the name “Cutthroat trout Oncorhynchus clarki”; on line 279 insert the term subsp. in normal font and add piscicida in italics as a subspecies. On line 280 there is a citation that escaped the authors not formatted correctly and not included in the references. Finally on line 286 put sp. in normal font.

The bibliography is poorly inserted: it is necessary to check the exact insertion of the citations bearing in mind that when multiple citations are inserted in the text, these must first be inserted in chronological order and eventually in alphabetical order when they are published in the same year; this principle then pours into the numbering of the references which are often incorrect. Furthermore, the insertion of bibliographic citations in the references must be complete both in the insertion of all authors correctly, and of the title and references to journals, with correct insertion of volume and relative pages.

For all the reasons stated above, I believe that this work does not have the right requirements to be published in the journal.

Author Response

Reviewer 2:

Suggestion: The manuscript vaccines-2013420 aims to track recent developments in vaccines against various bacterial pathogens affecting fish. Although written in a clear way, I do not find a linearity and a connection regarding the recent developments that have been conducted in recent years on this topic. This review lacks many ideas, especially original ones, to detach itself from the reviews already published in recent years in many scientific journals; I think it is not a suitable manuscript to be published, because it lacks novelty, ideas to become interesting.

Response: Here in this review article, we have discussed precisely over bacterial pathogens alone, as it will be easy to attract the people from various streams of academic research to industrial needs in a concise manner, where these data and references are accessible to the readers in short time.  Hence, we feel that it is to be good for publishing based on this requirement.

Suggestion

There are numerous inaccuracies in the text, such as line 105 where the species must be corrected.

In table 1 in the blot of Lactococcus garvieae, the rainbow trout is missing among the target species with the relative citations, as this gram-positive bacterium represents the main cause of death in this salmonid in farmed conditions. Streptococcus iniae, decidedly more important agent than S. parauberis, is also missing. It should be remembered that in line 133, the term spp is inserted in normal character. The pathology should be better highlighted on Moritella viscosa as it is among those most recently identified as a cause of pathology in fish. On line 183, italicize the scientific name and adjust the word Lancefield. On line 189 Mycobacterium goes in italics.

In Table 2 correct the name of disease Lactococcosis, correct the scientific name of the agent (Lactococcus garvieae), put spp. in normal font; insert ictaluri and ruckeri in lower case (last 2 lines) and correct the name of the etiological agent of Pasteurellosis.

 In line 247 correct the name “Cutthroat trout Oncorhynchus clarki”; on line 279 insert the term subsp. in normal font and add piscicida in italics as a subspecies. On line 280 there is a citation that escaped the authors not formatted correctly and not included in the references. Finally on line 286 put sp. in normal font.

Response:

Thank you for your extensive time taken for corrections based on the comments and suggestions all the changes have been made with the suggested “track changes” record of word tool from the document itself for your kind reference.

Round 2

Reviewer 1 Report

My previous (few) objections have been taken well care of by the authors in this new version of the manuscript. The new version is more global in the approach, showing that important progress is being done in many parts of the world.

I conclude that the authors have produced a useful review, which I hope will be read and taken into account by the fish farming industry globally. Vaccines are a key factor in our efforts to improve fish health and minimise use of antibacterial agents.

Reviewer 2 Report

The authors of the vaccine-2013420 manuscript fulfilled all the punctual requests of the reviewers. Based on these requirements, the paper could be published, not identifying other arguments to suggest.